# LO-BCQ: Locally Optimal Block Clustered Quantization for 4-bit (W4A4) LLM Inference

**Reena Elangovan**  *relangovan@nvidia.com*
*NVIDIA Corporation*

**Charbel Sakr**  *csakr@nvidia.com*
*NVIDIA Corporation*

**Anand Raghunathan**  *araghunathan@purdue.edu*
*Department of ECE*
*Purdue University*

**Brucek Khailany**  *bkhailany@nvidia.com*
*NVIDIA Corporation*

**Reviewed on OpenReview:** *https://openreview.net/forum?id=loWISTqGwW*

## Abstract

Post-training quantization (PTQ) is a promising approach to reducing the storage and computational requirements of large language models (LLMs) without additional training cost. Recent PTQ studies have primarily focused on quantizing only weights to sub-8-bits while maintaining activations at 8-bits or higher. Accurate sub-8-bit quantization for both weights and activations without relying on quantization-aware training remains a significant challenge. We propose a novel quantization method called block clustered quantization (BCQ) wherein each operand tensor is decomposed into blocks (a block is a group of contiguous scalars), blocks are clustered based on their statistics, and a dedicated optimal quantization codebook is designed for each cluster. As a specific embodiment of this approach, we propose a PTQ algorithm called Locally-Optimal BCQ (LO-BCQ) that iterates between the steps of block clustering and codebook design to greedily minimize the quantization mean squared error. When weight and activation scalars are encoded to W4A4 format (with 0.5-bits of overhead for storing scaling factors and codebook selectors), we advance the current state-of-the-art by demonstrating $< 1\%$ loss in inference accuracy across several LLMs and downstream tasks.

## 1 Introduction

Quantization is a highly effective and widely adopted technique for reducing the computational and storage demands of Large Language Model (LLM) inference. While recent efforts (Wang et al., 2023; Tseng et al., 2024; Egiazarian et al., 2024; Frantar et al., 2023; Lin et al., 2023) have largely focused on weight-only quantization targeting single-batch inference, activation quantization becomes critical for improving throughput during multi-batch inference scenarios such as cloud-scale deployments serving multiple users. Previous works (Yao et al., 2023; Dai et al., 2021) on sub-8-bit quantization of both weights and activations have relied on quantization-aware training (QAT) to recover accuracy loss during inference. However, the prohibitive cost of training and unavailability of training data in recent LLMs has made QAT increasingly difficult and motivated recent post-training quantization (PTQ) efforts (Xiao et al., 2023; Rouhani et al., 2023a; Wu et al., 2023).

In this paper, we develop a post-training quantization (PTQ) algorithm aimed at minimizing the mean squared error (MSE) of any operand tensor. Traditional MSE-optimal scalar quantization algorithms such

as Lloyd-Max (Lloyd, 1982) struggle to achieve aggressive bitwidth reduction without significant accuracy loss. To address this limitation, vector (or block) quantization methods have been explored in (Tseng et al., 2024; Egiazarian et al., 2024), which identify MSE-optimal vector codebooks. Despite their promising results for $\leq 4$-bit weight-only quantization, these existing approaches face two key challenges. First, they require complex codebook schemes involving weight updates to minimize quantization error, making them challenging to deploy for dynamic compression of activations. Second, they require large codebook sizes (on the order of $2^{16}$ codebooks with 8 entries each) per-model or per-layer to achieve aggressive model compression.

To overcome these limitations, we propose a novel clustering and quantization framework called block clustered quantization (BCQ). BCQ consists of two key steps:(1) a clustering step applied to operand blocks, and (2) a scalar quantization step individually applied to operand scalars based on their cluster membership. To minimize MSE in this process, we introduce LO-BCQ (locally optimal block clustered quantization), an iterative algorithm that jointly optimizes block clustering and per-cluster codebooks. We prove that LO-BCQ greedily minimizes quantization MSE across iterations by performing locally optimal steps at each iteration. We apply LO-BCQ on calibration data to identify optimal codebooks. We demonstrate that these codebooks can be frozen during inference across models and across linear layers within models. Using $\leq 16$ optimal codebooks with 16 entries each derived through LO-BCQ, we achieve state-of-the-art trade-offs between bitwidth and accuracy without requiring any weight updates during 4-bit quantization of both weights and activations across diverse models and downstream tasks.

## 1.1 Related work

Recent sub-4-bit quantization proposals such as (Wang et al., 2023; Tseng et al., 2024; Egiazarian et al., 2024) explore extreme weight quantization while maintaining activations at 8-bit or higher precision. In particular, BitNet (Wang et al., 2023) proposed W1A8 quantization resulting in an aggregate (weights + activations) bitwidth comparable to LO-BCQ. However, BitNet demands training from scratch and despite this large training cost suffers significant loss in accuracy in downstream tasks. QuiP# (Tseng et al., 2024) and AQLM (Egiazarian et al., 2024) propose W2A8 quantization through codebooks. These methods explore vector and additive codebook quantization, respectively, and rely on large codebook sizes (1MB of memory footprint) and are not directly applicable to activation quantization. These methods also require weight updates to minimize accuracy loss. In contrast, optimal codebooks ($\leq 0.19$KB of memory footprint) identified by LO-BCQ are applicable to both weight and activation quantization and achieves $< 1\%$ accuracy loss in downstream tasks without any weight updates. Supporting such small codebook sizes makes LO-BCQ more amenable to potential hardware acceleration of decompression. Minimizing quantization MSE using the 1D (Lloyd-Max) and 2D Kmeans clustering has been explored in (Han et al., 2016; Cho et al., 2021; 2023) and (van Baalen et al., 2024), respectively. In contrast, LO-BCQ iteratively optimizes

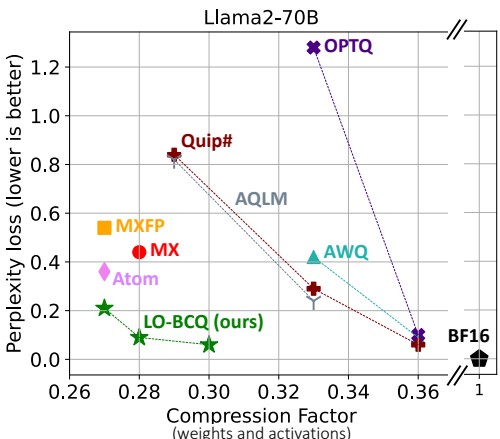

Figure 1: Wikitext perplexity loss relative to unquantized baseline vs compression factor of LO-BCQ compared to previous LLM quantization proposals. Here, compression factor is the cumulative number of bits in the weight and activation tensors[1] that need to processed in each layer relative to an unquantized BF16 baseline.

block clustering alongside Lloyd-Max based optimal scalar quantization of block clusters. W4A8 quantization has been proposed in (Frantar et al., 2023; Bai et al., 2021; Yao et al., 2022) involving weight updates to preserve accuracy and in (Lin et al., 2023; van Baalen et al., 2024) without any weight updates (PTQ). Further, (Guo et al., 2023; Wei et al., 2023; Kim et al., 2023a) perform sub-8-bit weight quantization by suppressing outliers.

Block (group) quantization is explored for aggressive quantization of both weights and activations in VSQ (Dai et al., 2021), FineQuant (Kim et al., 2023b), ZeroQuant-V2 (Yao et al., 2023), Atom (Zhao et al., 2024) through integer number formats, and in (Zhang et al., 2023), ZeroQuant-FP (Wu et al., 2023), MX (Rouhani et al., 2023a), MXFP (Rouhani et al., 2023b) and BSFP (Lo et al., 2023) through floating-point formats. Figure 1 compares the perplexity loss vs compression factor of LO-BCQ to other quantization proposals. Here, the perplexity loss is relative to an unquantized baseline on the Wikitext-103 dataset for LO-BCQ, MX and MXFP4, and on the Wikitext2 for others. The compression factor refers to the total number of bits in the weight and activation[1] tensors (computed as $|A|B_A + |W|B_W$ following Sakr et al. (2017))[2] that need to be processed in each layer relative to an unquantized BF16 baseline. Depending on the target application, weight or activation quantization may be more important. For the sake of generality, we consider them to be equally important in our metric. As shown in Figure 1, LO-BCQ advances the current state-of-the-art by achieving the best trade-off between perplexity and compression.

## 1.2 Contributions

The main contributions of this work are as follows:

- We propose BCQ, a block clustered quantization framework that performs per-block quantization by first clustering operand blocks and then quantizing each block cluster using a dedicated codebook.

- We derive a locally optimal version of BCQ called LO-BCQ that iteratively optimizes block clustering and per-cluster quantization to provably minimize quantization MSE for any value distribution. We demonstrate that LO-BCQ is applicable to quantization of both weights and activations of LLMs.

- We propose block formats for LO-BCQ where each operand block is associated with an index that maps it to one of a set of static codebooks, and a group of blocks (called a block array) share a quantization scale-factor. We vary the length of blocks, block arrays and the number of codebooks to study different configurations of LO-BCQ.

- When each of the weight and activation scalars are quantized to 4-bits (effective bitwidth including per-block scale-factors etc. is 4.5 to 4.625 bits), we achieve $< 0.1$ loss in perplexity across GPT3 (1.3B, 8B and 22B) and Llama2 (7B and 70B) models and $< 0.2$ loss in Nemotron4 (15B and 340B) models, respectively, on the Wikitext-103 dataset. Further, we achieve $< 1\%$ loss in average accuracy across downstream tasks such as MMLU and LM evaluation harness.

To the best of our knowledge, we are the first to achieve $< 1\%$ loss in downstream task accuracy when both LLM activations and weights are quantized to 4-bits during PTQ (no fine-tuning or weight updates).

## 2 Block Clustered Quantiaztion (BCQ)

In this section, we introduce the concept of block clustered quantization (BCQ) and present the locally optimal block clustered quantization (LO-BCQ) algorithm that minimizes quantization MSE for any operand. We also introduce block formats to support various LO-BCQ configurations.

### 2.1 Mathematical Definition

Given a tensor $\boldsymbol{X}$ composed of $L_X$ scalar elements, we denote its blockwise decomposition as $\{\boldsymbol{b}_i\}_{i=1}^{N_b}$, where $\boldsymbol{b}_i$'s are blocks of $L_b$ consecutive elements in $\boldsymbol{X}$, and the number of blocks is given by $N_b = L_X/L_b$. Block clustered quantization (see Figure 2) uses a family of $N_c$ codebooks $\mathcal{C} = \{C_i\}_{i=1}^{N_c}$, where $N_c << N_b$, and clusters the blocks into $N_c$ clusters such that each is associated with one of the $N_c$ codebooks. This procedure is equivalent to creating a mapping function $f$ from a block $b$ to a cluster index in $\{1, \ldots, N_c\}$. Quantization (or encoding) proceeds in a two-step process: (i) *mapping* to assign a cluster index to a given block, and (ii)

---

[1]. The size of activations is measured for the prefill phase with a context length of 4096 and batch size of 1.

[2] the notation $|X|$ refers to the total number of scalars in tensor $X$, and $B_X$ is the bitwidth of $X$.

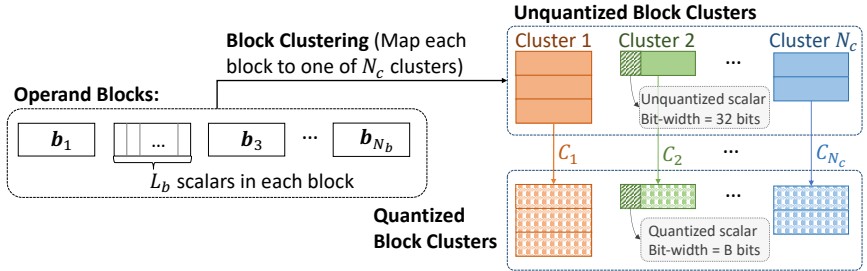

Figure 2: Block clustered quantization: Each operand block is first mapped to a cluster based on a mapping function and then each scalar of that block is encoded as a $B$-bit index to the closest entry in the $2^B$-entry codebook associated with that cluster.

*quantization* of its scalars using the codebook corresponding to that index. Formally, denoting $\hat{\boldsymbol{b}}$ as the result of block clustered quantization of a given block $\boldsymbol{b}$ in $\boldsymbol{X}$, this procedure is described as:

$$\hat{\boldsymbol{b}} = C_{f(\boldsymbol{b})}(\boldsymbol{b}) \tag{1}$$

where $C$ is a $2^B$-entry codebook that maps each scalar in $\boldsymbol{b}$ to a $B$-bit index to the closest representation. Each quantized scalar of block $\boldsymbol{b}$ is obtained as:

$$\hat{\boldsymbol{b}}[l] = \arg\min_{k=1\ldots 2^B} |\boldsymbol{b}[l] - C_{f(\boldsymbol{b})}[k]|^2 \tag{2}$$

where the notation $x[y]$ is used to describe the $y^{\text{th}}$ element in an arbitrary block $\boldsymbol{x}$. That is, each scalar in $\hat{\boldsymbol{b}}$ is an index to the closest entry by value in $C_{f(\boldsymbol{b})}$.

Once mapped by invoking $f$, we store the $log2(N_c)$-bit codebook selector for each block. Therefore, the effective bit-width of each quantized scalar is given by:

$$\text{Bitwidth}_{\text{BCQ}} = B + log2(N_c)/L_b \tag{3}$$

In practice, groups of blocks called a block-array share 8-bit scale factors each and the codebook entries are 6-bit integers. We discuss this further in section 2.4

## 2.2 Locally optimal block clustered quantization

Our goal is to construct a family of codebooks $\mathcal{C}$ resulting in minimal quantization MSE during block clustered quantization. Figure 3 presents an algorithm called Locally Optimal BCQ (LO-BCQ) to achieve this goal. LO-BCQ consists of two main steps: (i) updating block clusters with fixed per-cluster codebooks, and (ii) updating per-cluster codebooks with fixed block clusters. This algorithm begins at iteration 0 (initial condition) with a set of $N_c$ initial codebooks $\{C_1^{(0)}, \ldots, C_{N_c}^{(0)}\}$ and unquantized operand blocks as inputs. During step 1 of iteration $n$, with the per-cluster codebooks from the previous iteration $\{C_1^{(n-1)}, \ldots C_{N_c}^{(n-1)}\}$, we perform block clustering by mapping each block to the codebook that achieves minimum quantization MSE. That is, we use the following mapping function:

$$f^{(n)}(\boldsymbol{b}) = \arg\min_{i=1\ldots N_c} \|\boldsymbol{b} - C_i^{(n-1)}(\boldsymbol{b})\|_2^2 \tag{4}$$

Since each codebook $C_i$ is associated with a cluster $i$, mapping to $C_i$ is equivalent to mapping to cluster $i$. Specifically, at iteration $n$, we construct $N_c$ block clusters $\boldsymbol{\mathcal{B}}^{(n)} = \{\mathcal{B}_i^{(n)}\}_{i=1}^{N_c}$, where each cluster is defined as:

$$\mathcal{B}_i^{(n)} = \left\{ \boldsymbol{b}_j \big| f^{(n)}(\boldsymbol{b}_j) = i \text{ for } j \in \{1 \ldots N_b\} \right\} \tag{5}$$

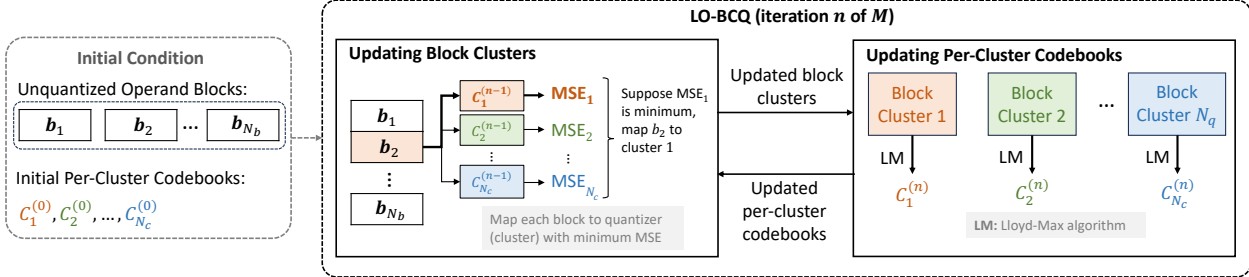

Figure 3: Overview of LO-BCQ algorithm: The algorithm starts with a set of initial per-cluster codebooks, and then iteratively performs two steps (i) fix per-cluster codebooks and update block clusters and (ii) fix block clusters and update per-cluster codebooks.

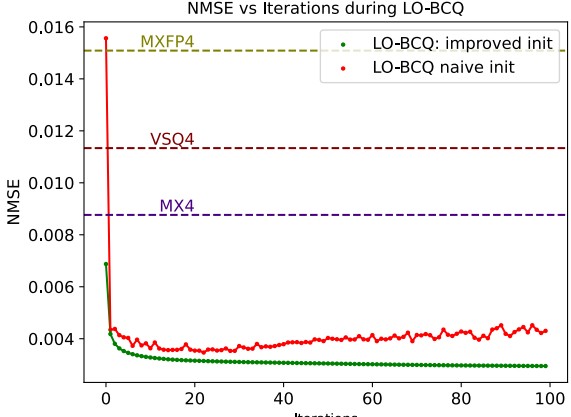

Figure 4: NMSE of LO-BCQ with naive initialization compared to the proposed initialization. Here LOBCQ is configured with a block array size of 64 and 16 codebooks.

In step 2, given the updated block clusters from step 1 and quantization bitwidth $B$, we apply the Lloyd-Max algorithm on each block cluster to derive optimal $2^B$-entry per-cluster codebooks $\{C_1^{(n)}, \ldots C_{N_c}^{(n)}\}$:

$$C_i^{(n)} \leftarrow \text{LloydMax}(\mathcal{B}_i^{(n)}, B) \tag{6}$$

where the Lloyd-Max algorithm (see A.1, Lloyd-Max is equivalent to K-means clustering on 1-dimensional data) is invoked on the data of the corresponding cluster $\mathcal{B}_i^{(n)}$.

We iterate steps 1 and 2 until convergence or a pre-determined number of iterations $M$. Empirically, we find that LO-BCQ converges at $M <= 100$. Since each of these steps are locally optimal, we find that the quantization MSE is non-increasing for each iteration. As a result, for any given value distribution, our LO-BCQ algorithm greedily minimizes quantization MSE. A theoretical proof of this claim is provided in section A.2.

## 2.3 Convergence and Initialization

Prior to clustering, we find that normalizing the operand blocks improves convergence. However, a block-wise normalization factor (or scaling factor) induces computational and memory overheads. Therefore, we perform a second-level quantization of this scaling factor to $B_s$-bits and share it across an array of blocks of length $L_A$. Furthermore, better convergence is observed for larger number of codebooks ($N_c$) and for a smaller block length ($L_b$). Such trends increase the bitwidth of BCQ in equation 3, meaning that LO-BCQ has an inherent trade-off between accuracy and complexity.

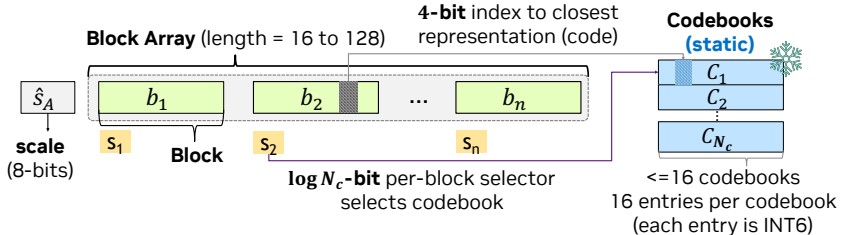

Figure 5: Block format for LO-BCQ. Each operand block is associated with a $log2(N_c)$-bit selector that selects the best codebook and each scalar is a 4-bit index that represents the closest value in the selected codebook. Each block array is associated with a 8-bit scale factor.

We initialize the per-cluster codebooks $\{C_1^{(0)}, \ldots, C_{N_c}^{(0)}\}$ based on K-means++ initialization algorithm which maximizes pairwise euclidean distances. In our experiments, we found such initialization to lead to significantly better convergence than a random one. Further, in step 2 of each iteration, when Lloyd-Max algorithm is invoked in equation 6, we set the initial centroids corresponding to the codebooks identified in the previous iteration. Figure 4 compares the MSE achieved by LO-BCQ with naive initialization to random codebooks to the proposed improved initialization. As shown, LO-BCQ with the proposed initialization converges to a lower MSE than the naive initialization and other block quantization baselines.

## 2.4 Block formats for LO-BCQ

Figure 5 illustrates the LO-BCQ block format where each operand block of length $L_b$ is associated with a $log2(N_c)$-bit index (result of the mapping function $f$ in equation 4) that selects the best codebook for that block. Each codebook is composed of $2^B$ entries and each scalar in the operand block is a $B$-bit index that represents the closest value in the selected codebook. Each entry in the codebook is a $B_c$-bit integer. Finally, each block array $\boldsymbol{A}$ is associated with a scale-factor $s_A$. This scale-factor and its quantization $\hat{s}_A$ to $B_s$-bits are computed as:

$$s_A = \left(2^{B_c - 1} - 1\right)/\max(\mathrm{abs}(\boldsymbol{A})) \tag{7}$$

$$\hat{s}_A = Q_F\left\{s_A/s_X, B_s\right\} \tag{8}$$

where $s_X$ is a per-tensor scale-factor that is shared by the entire operand tensor $\boldsymbol{X}$ and $Q_F$ denotes a quantizer that quantizes a given operand to format $F$ (see section A.4 for more details on number formats and quantization method).

The bitwidth of LO-BCQ is computed as:

$$\begin{aligned}
\mathrm{Bitwidth_{LO-BCQ}} = B + log2(N_c)/L_b + B_s/L_A \\
+ N_c * 2^B * B_c/L_X
\end{aligned} \tag{9}$$

where the term $N_c * 2^B * B_c/L_X$ is usually negligible since the memory footprint of codebooks (numerator) is negligible compared to the size of the operand tensor (denominator). Indeed, we emphasize that LO-BCQ shares a set of $N_c <= 16$ codebooks of size $<= 0.19KB$ among the scalars of the entire tensor, resulting in negligible memory overhead for storing the codebooks.

In this paper, we assume $B_s = 8$ and the data format $F$ is floating point E4M3. Further, each codebook entry is a 6-bit integer (i.e, $B_c = 6$) and we vary $N_c$ between 2 and 16, $L_b$ between 2 and 8, and $L_A$ between 16 and 128 to obtain various LO-BCQ configurations. We list the configurations and their corresponding bitwidths in Table 1. We perform a detailed ablation of these configurations and present our insights in section 4.3.

## 3 Applying LO-BCQ for LLM Inference

In this section, we discuss specifics of applying LO-BCQ for LLM inference. Specifically, we first describe the codebook design process, followed by method for activation quantization on-the-fly.

Table 1: Various LO-BCQ configurations and their bitwidths. Each block array of size $L_A$ share an 8-bit scale factor. Each operand block of size $L_b$ is associated with a $log2(N_c)$-bit index that maps it to one out of $N_c$ codebooks that best represents it.

| $N_c$ $L_A$ | $L_b = 8$ | | | | $L_b = 4$ | | $L_b = 2$ |
|---|---|---|---|---|---|---|---|
| | 2 | 4 | 8 | 16 | 2 | 4 | 2 |
| 128 | 4.1875 | 4.3125 | 4.4375 | 4.5625 | 4.3125 | 4.5625 | 4.5625 |
| 64 | 4.25 | 4.375 | 4.5 | 4.625 | 4.375 | 4.625 | 4.625 |
| 32 | 4.375 | 4.5 | 4.625 | 4.75 | 4.5 | 4.75 | 4.75 |
| 16 | 4.625 | 4.75 | 4.875 | 5 | 4.75 | 5 | 5 |

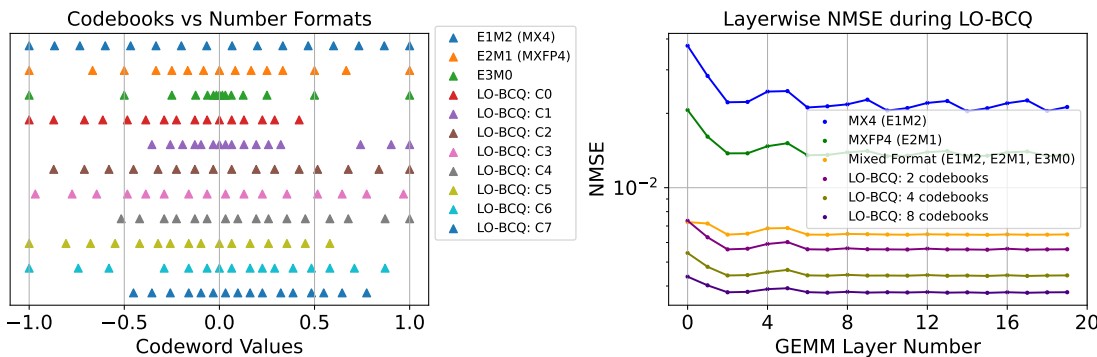

Figure 6: LO-BCQ codebooks compared to 4-bit floating point formats and layerwise normalized MSE (NMSE). We compute NMSE for the weights of first 20 GEMM layers (QKV, projection and fully-connected) of Llama2-7B model. Note that we use the NMSE for better visualization across varying layer data.

We pre-calibrate the LO-BCQ codebooks for both weights and activations offline (prior to inference). Since weights are known, their own data can be used as calibration set. On the other hand, activations are dynamic and vary for every input; thus, as per common quantization strategies (Wu et al., 2020a; Sakr et al., 2022), we employ a randomly sampled calibration set from training data in order to build activation codebooks. Once codebooks are calibrated, we also quantize the codewords to 6-bit integers. The choice of 6-bit was based on empirical observations of accuracy being maintained with $L_A <= 128$.

Figure 6 (left) compares the codebooks identified by the LO-BCQ algorithm in a GEMM layer of a GPT3-126M model to 4-bit floating point formats such as E1M2, E2M1 and E3M0. The LO-BCQ codebooks outperform other block formats as shown in Figure 6 (right) by capturing the arbitrary and non-uniform patterns in the value distributions of LLM operands and allowing each block to map to the codebook that best represents it. The mapping of operand blocks to the best of available codebooks can be conceptually compared to prior works that have explored mixed-format quantization such as (Tambe et al., 2020; Zadeh et al., 2022).

LO-BCQ provides the quantization operation the flexibility to assign data to any of the sign posts (codewords) in Figure 6(left). The union of these sign posts covers the real line with a resolution that is clearly superior to that of a 4-bit quantizer. Therefore, we hypothesized that these codebooks need not be calibrated on a per-tensor (layerwise) basis, but rather, it is likely that they would be universally appropriate to quantize *any tensor (weights and activations), at any layer, for any model*. To verify this hypothesis, we calibrated a set of codebooks on data sampled from GPT3 models on Wikitext-103 dataset and froze it. We find that these codebooks achieve comparable quantization MSE compared to those calibrated individually on each operand as shown in Figure 7 which verifies our hypothesis. In our subsequent results, we always employ universally calibrated codebooks.

Finally, the small size of LO-BCQ codebooks enables efficient activation quantization on the fly. Indeed, LO-BCQ involves computing the following values – per-block array scale-factor $s_A$, per-block codebook

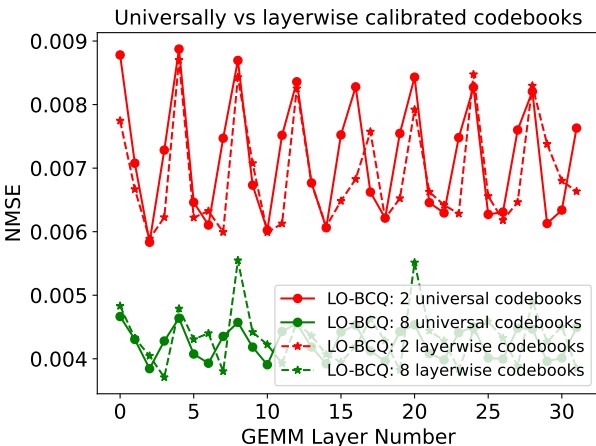

Figure 7: Quantization NMSE acheived by universally calibrated codebooks compared to that calibrated layerwise in Llama2-7B inputs of first 30 GEMM (QKV, projection and fully-connected) layers.

selector $s_b$ which is the result of the mapping function $f$ (Eq. 4), and the index to closest representation $\hat{b}$ in the selected codebook (Eq. 2). Note that the computation of $s_A$, similar to other block quantization methods (Rouhani et al., 2023b; Dai et al., 2021), simply corresponds to a max-reduction (followed by quantization) over the block array whose size is small ($<= 128$). Further, with LO-BCQ, the codebooks are constant (frozen) with small size ($<= 0.19KB$). This is an important distinction with other works on codebook quantization (Tseng et al., 2024; Egiazarian et al., 2024). As such, $s_b$ and $\hat{b}$ can be concurrently computed across operand blocks.

# 4 Experimental Evaluation of LO-BCQ

In this section, we present our accuracy studies on downstream tasks comparing LO-BCQ to various other block quantization proposals. Next, we present ablation studies on varying LO-BCQ configurations and our calibration methodology, namely universal vs local.

## 4.1 Experimental Setup

We perform accuracy studies on GPT3 (Shoeybi et al., 2020) (1.3B, 8B and 22B), Llama2 (Touvron et al., 2023) (7B and 70B) and Nemotron4 (15B and 340B) (Parmar et al., 2024) models. We evaluate PTQ inference accuracy on several downstream tasks including Wikitext-103 (Merity et al., 2016), MMLU (Hendrycks et al., 2021) and Eleuther AI's LM evaluation harness (Gao et al., 2024). In LM evaluation harness, we infer on Race (RA), Boolq (BQ), Hellaswag (HS), Piqa (PQ) and Winogrande (WG) tasks and in the MMLU dataset we evaluate all tasks. In all these models, we quantize GEMM layers including Query, Key and Value computations, Projection layer after self attention and the fully-connected layers.

We apply the LO-BCQ algorithm to the operands before inference and pre-calibrate the optimal codebooks. In our experiments, we perform this calibration on one batch of activations from the training data of the GPT3-126M model and the Wikitext-103 dataset. We freeze these optimal codebooks across operands and models during all of our accuracy evaluations. Further, we represent each entry of the codebooks as a 6-bit integer. That is, once decoded, the inner product computations with a block array during inference can be performed at 6-bit precision[3]. Furthermore, we perform ablation studies on the LO-BCQ configurations with quantization bitwidth ranging from 4.25-bits to 5-bits.

We compare LO-BCQ against previous block quantization works that have explored PTQ of both weights and activations such as VSQ (Dai et al., 2021), MX (Rouhani et al., 2023a), MXFP (Rouhani et al., 2023b),

---

[3]In our experiments in this paper, we emulate ("fake") quantization by representing the quantized values in BF16 format. Therefore, the computations are performed in BF16 precision.

Table 2: PTQ Perplexity (lower is better) on Wikitext-103 dataset with GPT3, Llama2 and Nemotron4 models.

| Method | Bitwidth* (W4A4) | Wikitext-103 PPL ($\Delta$) | | | | | |
| --- | --- | --- | --- | --- | --- | --- | --- |
| | | GPT3 | | Llama2 | | Nemotron4 | |
| | | 8B | 22B | 7B | 70B | 15B | 340B |
| BF16 (Pretrained) | 16 | 7.38 | 6.54 | 5.06 | 3.14 | 5.87 | 3.48 |
| MX4 (g16) | 4.5 | 8.15 (0.77) | 7.69 (1.15) | 5.73 (0.67) | 3.58 (0.44) | 8.88 (3.01) | 4.01 (0.53) |
| VSQ (g16) | 4.5 | 8.17 (0.79) | 7.12 (0.58) | 835 (829) | 4.96 (1.82) | 7.58 (1.71) | 4.19 (0.71) |
| MXFP4 (g32) | 4.25 | 9.12 (1.74) | 10.18 (3.64) | 5.76 (0.70) | 3.69 (0.55) | 8.24 (2.37) | 4.10 (0.62) |
| **LO-BCQ (g64, $N_c = 2$)** | 4.25 | 7.61 (0.23) | 6.74 (0.20) | 5.31 (0.25) | 3.35 (0.21) | 6.30 (0.43) | 3.67 (0.19) |
| **LO-BCQ (g64, $N_c = 8$)** | 4.5 | **7.48 (0.10)** | **6.62 (0.08)** | 5.19 (0.13) | **3.23 (0.09)** | 6.13 (0.26) | 3.60 (0.12) |
| **LO-BCQ (g32, $N_c = 16$)** | 4.75 | **7.45 (0.07)** | **6.59 (0.05)** | **5.15 (0.09)** | **3.20 (0.06)** | 6.03 (0.16) | **3.56 (0.08)** |

\* Bitwidth of weights and activations including the overheads from per-block array (group) scale and codebook selectors

QuaRot (Ashkboos et al., 2024), Atom (Zhao et al., 2024), OmniQuant (Shao et al., 2024) and SmoothQuant (Xiao et al., 2023). VSQ and MX perform per-block quantization of 16-element blocks with an 8-bit scale-factor per-block resulting in an effective bit-width of 4.5 bits. VSQ quantizes each scalar to INT4 format and per-block scale-factor to INT8 format. MX performs micro-scaling at per-block level with a 1-bit exponent shared by 2-element blocks. Each scalar is quantized to INT3. In this paper, we overestimate accuracy of MX by allowing each scalar to have its own exponent, resulting in INT4 precision. The per-block array scale factors of MX are quantized to E8M0 format. Therefore, our evaluation results in a bitwidth of 4.5 bits. Further, MXFP explores 32-element blocks with 8-bit scale-factor per block resulting in an effective bitwidth of 4.25 bits. The number format of scalars and per-block scale factors are E2M1 and E8M0, respectively. The quantization methodology with these block formats is detailed in A.4.3.

Additionally, we compare weight-only (W4A8) LO-BCQ to other weight-only quantization proposals of equivalent bitwidth such as GPTQ (Frantar et al., 2023), AWQ (Lin et al., 2023), QuiP# (Tseng et al., 2024) and AQLM (Egiazarian et al., 2024). For this comparison, we choose a block-array length of 128 for LO-BCQ, matching the group-size of other works.

## 4.2 Accuracy studies on downstream tasks

We present our comprehensive accuracy evaluations across the Nemotron4, Llama2 and GPT3 models, on the Wikitext-103, LM evaluation harness and MMLU datasets. For convenience, we present select LO-BCQ configurations with $L_b = 8$ in this section. See **??** for accuracy studies on other configurations.

### 4.2.1 Perplexity on Wikitext-103

As shown in Table 2, across large models such as Llama2-70B, Nemotron4-340B and GPT3-22B, 4.5-bit LO-BCQ achieves $\leq 0.12$ loss in perplexity compared to the unquantized baseline on the Wikitext-103 dataset. MX, MXFP and VSQ perform per-block quantization by associating a scale-factor to each block (or a block array) and with a single number format (quantizer) across blocks. On the other hand, in addition to per-block array scaling, LO-BCQ allows a block to flexibly map to a codebook that best represents it from a set of codebooks. This flexibility allows LO-BCQ to achieve better perplexity. Furthermore, we find that with a larger quantization bitwidth, LO-BCQ achieves better perplexity across models as expected. We achieve these improvements during PTQ, i.e., without any additional training or finetuning.

The number format of per-group (or block array) scale-factor has a significant impact on accuracy. VSQ is unable to sufficiently capture the range of activations with its INT8 scale-factors as observed in Llama2-7B, while it outperforms the E8M0 scale-factors of MX in GPT3-22B due to better resolution when representing large values. Across various models, we find that the E4M3 format of LO-BCQ provides sufficient range and resolution to represent the scale-factors.

Table 3 compares LO-BCQ to other PTQ methods that quantize both weights and activations. Here, all methods use a group (block array) size of 128. As shown, LO-BCQ significantly outperforms the prior-art. While the prior-art proposes various techniques for suppressing outliers in both weights and activations, LO-BCQ calibrates a set of codebooks that captures various non-uniform value distributions.

Table 3: Comparing perplexity loss (lower is better) of LO-BCQ to other 4-bit (W4A4) quantization works such as QuaRot, Atom, OmniQuant and SmoothQuant. Here, the perplexity loss is on Wikitext-103 dataset for LO-BCQ and Wiki2 for others.

| Method | Bitwidth | $\Delta$ Wiki PPL | |
|---|---|---|---|
| (g128) | (W4A4) | Llama2-7B | Llama2-70B |
| SmoothQuant | 4.13 | 77.65 | - |
| OmniQuant | 4.13 | 9.14 | - |
| QuaRot | 4.13 | 0.46 | 0.29 |
| Atom | 4.13 | 0.56 | 0.36 |
| **LO-BCQ** ($N_c = 2$) | 4.19 | 0.14 | **0.09** |
| **LO-BCQ** ($N_c = 4$) | 4.31 | 0.12 | **0.07** |
| **LO-BCQ** ($N_c = 8$) | 4.44 | **0.09** | **0.06** |
| **LO-BCQ** ($N_c = 16$) | 4.56 | **0.08** | **0.05** |

Table 4: Comparing perplexity loss of weight-only (W4A16) LO-BCQ to other weight-only quantization methods.

| Method | Bitwidth | #codebooks | Llama2-7B | | | | Llama2-70B | | | |
|---|---|---|---|---|---|---|---|---|---|---|
| | (W4A16) | | $\Delta$ Wiki PPL | PQ | WG | HS | $\Delta$ Wiki PPL | PQ | WG | HS |
| GPTQ (g128) | 4 | - | 0.37 | 76.61 | 68.19 | 55.44 | 0.23 | 81.23 | 75.61 | 63.47 |
| AWQ (g128) | 4.13 | - | 0.13 | 77.09 | **69.53** | 56.25 | **0.09** | - | - | - |
| QuiP# | 4.02 | $2 * 2^{16}$ | 0.17 | 77.91 | 66.85 | 55.78 | 0.10 | 81.45 | 76.8 | 63.51 |
| AQLM | 4.14 | $2 * 2^{16}$ | **0.09** | 78.24 | 67.32 | 55.99 | **0.07** | 81.5 | 76.48 | 63.69 |
| **LO-BCQ (g128)** | 4.19 | 2 | 0.14 | **78.29** | 68.11 | **56.55** | 0.09 | 81.18 | **79.24** | **64.57** |
| | 4.31 | 4 | 0.12 | 78.18 | 68.59 | **56.75** | 0.07 | **81.77** | 78.30 | **64.99** |
| | 4.44 | 8 | **0.09** | 77.69 | 68.75 | **56.75** | 0.06 | 81.50 | **79.79** | **65.11** |
| | 4.56 | 16 | **0.08** | 77.97 | 68.90 | **56.76** | 0.05 | 81.45 | **80.43** | **65.04** |

Table 5: Comparing perplexity loss of sub-4-bit weight-only LO-BCQ to other weight-only quantization methods.

| | Method | Bitwidth | #codebooks | Wiki2 PPL | | |
|---|---|---|---|---|---|---|
| | | | | Llama2-7B | Llama2-13B | Llama2-70B |
| | BF16 (Pretrained) | 16 | - | 5.47 | 4.88 | 3.32 |
| W3A16 | QuiP# (LDLQ, no FT) | 3 | $2^{16} + 2^8$ | 5.91 | 5.23 | 3.61 |
| | AQLM (FT) | 3 | $2 * 2^{12}$ | 5.46 | 4.82 | 3.36 |
| | LO-BCQ (LDLQ, no FT) | 3.375 | 4 | 5.79 | 5.12 | 3.53 |
| | | 3.5 | 8 | 5.72 | 5.09 | 3.49 |
| W2A16 | QuiP# (LDLQ, no FT) | 2 | $2^{16}$ | 8.05 | 6.59 | 4.44 |
| | AQLM (FT) | 2 | $2^{16}$ | 6.59 | 5.60 | 3.94 |
| | LO-BCQ (LDLQ, no FT) | 2.375 | 4 | 8.02 | 7.12 | 4.52 |
| | | 2.5 | 8 | 6.87 | 6.16 | 4.20 |

### 4.2.2 Accuracy on LM evaluation harness tasks

Across 0-shot LM evaluation harness tasks in Table 6, LO-BCQ shows significant improvement in average accuracy compared to MX, MXFP and VSQ at equivalent bitwidth. Further, across models during 4.5-bit quantization, LO-BCQ achieves $< 1\%$ loss in average accuracy compared to the respective unquantized baselines. When the bitwidth of LO-BCQ is increased by varying its configuration, we find that the average accuracy generally increases albeit with a few exceptions. Although these variations are small ($< 0.5\%$), we believe that they arise due to the universal calibration of codebooks. Our codebooks are calibrated on a batch of training data from the Wikitext-103 dataset and the GPT3-126M model and remain frozen across all datasets and models.

Although the focus of this work is weight+activation quantization, we also compare to prior art weight-only quantization proposals for completeness. Table 4 compares weight-only (W4A8) LO-BCQ with a block array size of 128 to other weight-only quantization proposals such as GPTQ and AWQ of comparable block array size and effective bit-width. As shown, LO-BCQ with 2, 4, 8 and 16 codebooks with effective bitwidth of 4.19, 4.31, 4.44 and 4.56, respectively, achieves significantly lower perplexity loss. It is worth noting that we evaluate this loss on Wikitext-103 dataset, which is a much larger dataset compared to Wikitext2 used by other works. Further, we compare LO-BCQ against other codebook-based quantization methods such as Quip# and AQLM. As shown, LO-BCQ achieves comparable accuracy using only a small codebook size (8 codebooks with 16 entries each) compared to significantly larger codebook sizes ($2^{16}$ codebooks with 8 entries each) required by QuiP# and AQLM.

Table 6: LM evaluation Harness 0-shot accuracy (higher is better) on Llama2 and GPT3 models.

| Method | Bitwidth (W4A4) | Llama2-7B | | | | | |
|---|---|---|---|---|---|---|---|
| | | RA | BQ | WG | PQ | HS | Avg (Δ %) |
| BF16 | 16 | 44.4 | 79.29 | 69.38 | 78.07 | 57.10 | 65.65 |
| QuaRot (g128) | 4.13 | - | - | 63.77 | 76.77 | - | - |
| MX4 (g16) | 4.5 | 41.43 | 73.98 | 66.22 | 77.04 | 55.19 | 62.77 (2.88) |
| VSQ (g16) | 4.5 | 31.39 | 65.75 | 55.49 | 67.30 | 43.51 | 52.69 (12.96) |
| MXFP4 (g32) | 4.25 | 41.34 | 74.00 | 67.48 | **77.53** | 54.22 | 62.91 (2.74) |
| **LO-BCQ (g64, $N_c = 2$)** | 4.25 | 42.49 | 77.58 | **68.90** | 77.09 | 55.93 | **64.40 (1.25)** |
| **LO-BCQ (g64, $N_c = 8$)** | 4.5 | 42.58 | 77.43 | **69.77** | 77.09 | 56.51 | **64.68 (0.97)** |
| **LO-BCQ (g32, $N_c = 16$)** | 4.75 | **43.73** | 77.86 | 68.90 | 77.86 | **56.52** | **64.97 (0.68)** |
| | | Llama2-70B | | | | | |
| BF16 | 16 | 48.8 | 85.23 | 79.95 | 81.56 | 65.27 | 72.16 |
| QuaRot (g128) | 4.13 | - | - | 76.24 | 82.43 | - | - |
| MX4 (g16) | 4.5 | **48.04** | 82.41 | 76.40 | **80.58** | 63.24 | 70.13 (2.03) |
| VSQ (g16) | 4.5 | 47.85 | 82.29 | 77.27 | 79.82 | 61.40 | 69.73 (2.43) |
| MXFP4 (g32) | 4.25 | 47.75 | 83.06 | 76.32 | **80.58** | 63.24 | 70.19 (1.97) |
| **LO-BCQ (g64, $N_c = 2$)** | 4.25 | **49.0** | 82.82 | 78.77 | 81.45 | 64.21 | **71.25 (0.91)** |
| **LO-BCQ (g64, $N_c = 8$)** | 4.5 | **49.28** | 84.03 | 78.37 | 81.45 | **64.76** | **71.58 (0.58)** |
| **LO-BCQ (g32, $N_c = 16$)** | 4.75 | **49.28** | **84.93** | 80.66 | 81.34 | 65.18 | **72.28 (+0.12)** |
| | | GPT3-8B | | | | | |
| BF16 | 16 | 41.34 | 68.32 | 67.88 | 78.78 | 54.16 | 62.10 |
| MX4 (g16) | 4.5 | 38.28 | 66.27 | 65.11 | 75.63 | 50.77 | 59.21 (2.89) |
| VSQ (g16) | 4.5 | **40.86** | 63.91 | 66.93 | 76.28 | 51.38 | 59.87 (2.23) |
| MXFP4 (g32) | 4.25 | 39.71 | 65.35 | 67.01 | 76.12 | 50.22 | 59.68 (2.42) |
| **LO-BCQ (g64, $N_c = 2$)** | 4.25 | **40.48** | 69.20 | 66.85 | 77.31 | 53.06 | **61.38 (0.72)** |
| **LO-BCQ (g64, $N_c = 8$)** | 4.5 | 39.43 | **69.45** | **67.72** | **77.75** | **53.71** | **61.61 (0.49)** |
| **LO-BCQ (g32, $N_c = 16$)** | 4.75 | 39.62 | 69.30 | 67.00 | 77.37 | **53.51** | **61.36 (0.74)** |
| | | GPT3-22B | | | | | |
| BF16 | 16 | 40.67 | 76.54 | 70.64 | 79.16 | 57.11 | 64.82 |
| MX4 (g16) | 4.5 | 39.04 | 72.26 | 67.96 | 77.86 | 54.77 | 62.38 (2.44) |
| VSQ (g16) | 4.5 | **40.57** | 65.81 | **69.61** | 77.20 | 54.82 | 61.60 (3.22) |
| MXFP4 (g32) | 4.25 | 39.14 | 69.61 | 64.17 | 75.68 | 47.60 | 59.24 (5.58) |
| **LO-BCQ (g64, $N_c = 2$)** | 4.25 | **40.48** | 75.41 | 69.14 | **78.24** | 56.06 | **63.87 (0.95)** |
| **LO-BCQ (g64, $N_c = 8$)** | 4.5 | 39.43 | **77.09** | **70.17** | 78.62 | **56.60** | **64.38 (0.44)** |
| **LO-BCQ (g32, $N_c = 16$)** | 4.75 | **39.62** | 75.35 | 69.30 | **78.89** | **56.64** | **63.96 (0.86)** |

Table 7: MMLU accuracy (higher is better) with Nemotron4-15B, Llama2-7B, 70B and GPT3-22B models.

| Method | Bitwidth (W4A4) | Nemo4 15B | Llama2 7B | Llama2 70B | GPT3 22B |
|---|---|---|---|---|---|
| BF16 | 16 | 64.3 | 45.8 | 69.12 | 38.75 |
| MX4 (g16) | 4.5 | 58.15 | 41.38 | 65.73 | 37.07 |
| VSQ (g16) | 4.5 | 57.38 | 26.48 | 62.46 | 37.79 |
| MXFP4 (g32) | 4.25 | 58.28 | 37.64 | 66.16 | 32.26 |
| **LO-BCQ (g64, $N_c = 2$)** | 4.25 | 63.17 | 43.90 | 68.07 | 36.71 |
| **LO-BCQ (g64, $N_c = 8$)** | 4.5 | **63.72** | 43.90 | 68.17 | 38.13 |
| **LO-BCQ (g32, $N_c = 16$)** | 4.75 | **64.33** | 44.50 | 68.27 | 38.34 |

Table 8: Perplexity on Wikitext-103 dataset across various LO-BCQ configurations

| $L_b \rightarrow$ | 8 | | | | 4 | | 2 |
|---|---|---|---|---|---|---|---|
| $N_c$ / $L_A$ | 2 | 4 | 8 | 16 | 2 | 4 | 2 |
| **Llama2-70B (FP32 PPL = 3.14)** | | | | | | | |
| 64 | 3.35 | 3.25 | 3.23 | 3.21 | 3.31 | 3.22 | 3.27 |
| 32 | 3.27 | 3.24 | 3.22 | 3.20 | 3.25 | 3.22 | 3.22 |
| 16 | 3.25 | 3.22 | 3.20 | 3.19 | 3.23 | 3.20 | 3.20 |
| **GPT3-22B (FP32 PPL = 6.54)** | | | | | | | |
| 64 | 6.74 | 6.64 | 6.62 | 6.63 | 6.71 | 6.64 | 6.64 |
| 32 | 6.67 | 6.64 | 6.61 | 6.59 | 6.65 | 6.64 | 6.60 |
| 16 | 6.67 | 6.63 | 6.59 | 6.61 | 6.66 | 6.63 | 6.62 |

Table 9: Perplexity on Wikitext-103 dataset with universally calibrated vs locally calibrated codebooks

Llama2-7B (FP32 PPL = 5.06), $L_b = 8$

| $N_c$ / $L_A$ | 2 | 4 | 8 | 16 | 2 | 4 | 8 | 16 |
|---|---|---|---|---|---|---|---|---|
| | Universally Calibrated Codebooks | | | | Layerwise Calibrated Codebooks | | | |
| 64 | 5.31 | 5.26 | 5.19 | 5.18 | 5.29 | 5.22 | 5.19 | 5.17 |
| 32 | 5.23 | 5.25 | 5.18 | 5.15 | 5.23 | 5.19 | 5.17 | 5.15 |
| 16 | 5.23 | 5.19 | 5.16 | 5.14 | 5.20 | 5.17 | 5.15 | 5.14 |

Table 10: Quantizing LO-BCQ codebook entries to INT4 vs INT6 vs INT8 on Wikitext-103. Perplexity is measured on the Llama2-7B model (BF16 PPL = 5.06).

| Method | Bitwidth of codewords | | |
|---|---|---|---|
| | INT4 | INT6 | INT8 |
| **LO-BCQ (g128, $N_c = 2$)** | 6.24 | 5.38 | 5.35 |
| **LO-BCQ (g128, $N_c = 4$)** | 6.21 | 5.27 | 5.25 |
| **LO-BCQ (g128, $N_c = 8$)** | 6.20 | 5.21 | 5.21 |
| **LO-BCQ (g128, $N_c = 16$)** | 6.18 | 5.21 | 5.19 |

Table 5 shows the Wiki2 perplexity results on weight-only quantization of Llama2 models with LO-BCQ and compares against state-of-the-art 2-bit and 3-bit quantization proposals such as QuIP# and AQLM. We find that despite the significantly smaller number of codebooks ($<= 8$ with 16 entries each) utilized by LO-BCQ compared to about $2^{12}$ to $2^{16}$ codebooks with 8 entries each in the other methods, LO-BCQ achieves competitive results. During 3-bit quantization, LO-BCQ with effective bitwidth of 3.375 bits achieves lower perplexity than QuIP# where LDLQ Tseng et al. (2024) is applied to both methods and no finetuning is performed. Similarly, when compared against 2-bit QuIP#, we find that LO-BCQ with effective bitwidth of 2.375 bits achieves better Wiki2 perplexity. Further, 3.5-bit and 2.5-bit LO-BCQ without finetuning suffers only a small loss compared to finetuned AQLM during 3-bit and 2-bit quantization of weights, respectively.

### 4.2.3 Accuracy on MMLU tasks

Similarly, in 5-shot MMLU tasks LO-BCQ achieves $< 1\%$ loss in average accuracy with 4.5-bits per scalar compared to respective unquantized baselines across GPT3-22B and Llama2-70B models. Further, LO-BCQ achieves a significantly better accuracy compared to all of our block quantization baselines such as VSQ, MX and MXFP4 at equivalent bitwidth. Across Llama2 models, LO-BCQ with a smaller bitwidth (4.25-bits) outperforms VSQ and MX4 with a comparatively larger bitwidth (4.5-bits). While the 0.5-bit overhead in VSQ and MX4 are used on per-block array scale-factors, the 0.25-bit overhead of LO-BCQ is shared between scale-factors and codebook selectors. Therefore, the superior accuracy of LO-BCQ can be attributed to the better representation by selecting the best codebook for each block.

### 4.3 Ablation Studies

Table 8 shows the perplexity of LO-BCQ on Wikitext-103 dataset and across Llama2-70B and GPT3-22B models when its configuration is varied. For a given $L_b$ (block length), larger number of codebooks results in better perplexity. This is intuitive since larger number of codebooks leads to better representation of the values in each block since LO-BCQ allows it to map to the codebook with best representation. Further, when the block array size is reduced, we achieve better perplexity. The block array corresponds to the granularity of normalization. As discussed in section 2.3, normalization improves convergence of LO-BCQ and results in better perplexity. When comparing configurations with same bitwidth, we find that the configuration with larger number of codebooks is better than smaller block array. This shows that the per-block metadata is better utilized for codebook selectors than scale factors. Furthermore, we find that reducing the block length ($L_b$) below 8 results in diminishing returns. This is because, the overhead of storing codebook selectors is larger for a smaller block. For a given bitwidth, configuration with smaller $L_b$ has fewer codebooks. Therefore, these configurations result in larger loss in perplexity. Table 10 compares perplexity achieved by LO-BCQ codebooks with INT4, INT6 and INT8 entries on Llama2-7B model and Wikitext-103 datatset. As shown, LO-BCQ with INT6 achieves negligible perplexity degradation compared to INT8, while INT4 suffers significantly larger degradation. As a result, we quantize LO-BCQ codebooks to INT6.

Table 9 compares the perplexity with universally calibrated codebooks to codebooks calibrated layerwise (per-tensor) in Llama2-7B model. The layerwise calibrated codebooks achieve slightly better perplexity when the number of codebooks are small (e.g. $N_c = 2$). However, they do not provide significant benefits when $N_c > 4$ despite the comparatively larger calibration effort. Therefore, in our experiments in this paper, we have largely explored universally calibrated codebooks.

# 5 Conclusion and Future Work

We propose a new iterative block clustering and quantization algorithm called LO-BCQ, that greedily minimizes quantization MSE for any operand (weights and activations) through locally optimal steps at each step of the iteration. We demonstrate that LO-BCQ achieves state-of-the-art perplexity across a suite of GPT3, LLama2 and Nemotron4 models on various downstream tasks. Given its strong inference accuracy with W4A4 quantization, we believe LO-BCQ opens new research avenues for even more aggressive quantization of both weights and activations. Furthermore, our approach requires significantly fewer codebooks than prior codebook-based methods and allows these codebooks to be static (frozen) across models and layers within models. This creates new opportunities to improve inference efficiency, which we plan to explore in future work.

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

# A Appendix

## A.1 Lloyd-Max Algorithm

For a given quantization bitwidth $B$ and an operand $\boldsymbol{X}$, the Lloyd-Max algorithm finds $2^B$ quantization levels $\{\hat{x}_i\}_{i=1}^{2^B}$ such that quantizing $\boldsymbol{X}$ by rounding each scalar in $\boldsymbol{X}$ to the nearest quantization level minimizes the quantization MSE.

The algorithm starts with an initial guess of quantization levels and then iteratively computes quantization thresholds $\{\tau_i\}_{i=1}^{2^B-1}$ and updates quantization levels $\{\hat{x}_i\}_{i=1}^{2^B}$. Specifically, at iteration $n$, thresholds are set to the midpoints of the previous iteration's levels:

$$\tau_i^{(n)} = \frac{\hat{x}_i^{(n-1)} + \hat{x}_{i+1}^{(n-1)}}{2} \text{ for } i = 1 \dots 2^B - 1$$

Subsequently, the quantization levels are re-computed as conditional means of the data regions defined by the new thresholds:

$$\hat{x}_i^{(n)} = \mathbb{E}\left[\boldsymbol{X} \big| \boldsymbol{X} \in [\tau_{i-1}^{(n)}, \tau_i^{(n)}]\right] \text{ for } i = 1 \dots 2^B$$

where to satisfy boundary conditions we have $\tau_0 = -\infty$ and $\tau_{2^B} = \infty$. The algorithm iterates the above steps until convergence.

Figure 8 compares the quantization levels of a 7-bit floating point (E3M3) quantizer (left) to a 7-bit Lloyd-Max quantizer (right) when quantizing a layer of weights from the GPT3-126M model at a per-tensor granularity. As shown, the Lloyd-Max quantizer achieves substantially lower quantization MSE. Further, Table 11 shows the superior perplexity achieved by Lloyd-Max quantizers for bitwidths of 7, 6 and 5. The difference between the quantizers is clear at 5 bits, where per-tensor FP quantization incurs a drastic and unacceptable increase in perplexity, while Lloyd-Max quantization incurs a much smaller increase. Nevertheless, we note that even the optimal Lloyd-Max quantizer incurs a notable ($\sim 1.5$) increase in perplexity due to the coarse granularity of quantization.

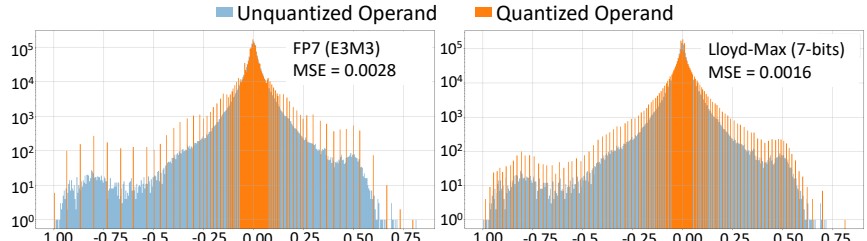

Figure 8: Quantization levels and the corresponding quantization MSE of Floating Point (left) vs Lloyd-Max (right) Quantizers for a layer of weights in the GPT3-126M model.

Table 11: Comparing perplexity (lower is better) achieved by floating point quantizers and Lloyd-Max quantizers on a GPT3-126M model for the Wikitext-103 dataset.

| Bitwidth | Floating-Point Quantizer | | Lloyd-Max Quantizer |
|---|---|---|---|
| | Best Format | Wikitext-103 Perplexity | Wikitext-103 Perplexity |
| 7 | E3M3 | 18.32 | 18.27 |
| 6 | E3M2 | 19.07 | 18.51 |
| 5 | E4M0 | 43.89 | 19.71 |

## A.2 Proof of Local Optimality of LO-BCQ

For a given block $\boldsymbol{b}_j$, the quantization MSE during LO-BCQ can be empirically evaluated as $\frac{1}{L_b}\|\boldsymbol{b}_j - \hat{\boldsymbol{b}}_j\|_2^2$ where $\hat{\boldsymbol{b}}_j$ is computed from equation (1) as $C_{f(\boldsymbol{b}_j)}(\boldsymbol{b}_j)$. Further, for a given block cluster $\mathcal{B}_i$, we compute

the quantization MSE as $\frac{1}{|\mathcal{B}_i|}\sum_{\boldsymbol{b}\in\mathcal{B}_i}\frac{1}{L_b}\|\boldsymbol{b}-C_i^{(n)}(\boldsymbol{b})\|_2^2$. Therefore, at the end of iteration $n$, we evaluate the overall quantization MSE $J^{(n)}$ for a given operand $\boldsymbol{X}$ composed of $N_c$ block clusters as:

$$J^{(n)} = \frac{1}{N_c}\sum_{i=1}^{N_c}\frac{1}{|\mathcal{B}_i^{(n)}|}\sum_{\boldsymbol{v}\in\mathcal{B}_i^{(n)}}\frac{1}{L_b}\|\boldsymbol{b}-B_i^{(n)}(\boldsymbol{b})\|_2^2$$

At the end of iteration $n$, the codebooks are updated from $\mathcal{C}^{(n-1)}$ to $\mathcal{C}^{(n)}$. However, the mapping of a given vector $\boldsymbol{b}_j$ to quantizers $\mathcal{C}^{(n)}$ remains as $f^{(n)}(\boldsymbol{b}_j)$. At the next iteration, during the vector clustering step, $f^{(n+1)}(\boldsymbol{b}_j)$ finds new mapping of $\boldsymbol{b}_j$ to updated codebooks $\mathcal{C}^{(n)}$ such that the quantization MSE over the candidate codebooks is minimized. Therefore, we obtain the following result for $\boldsymbol{b}_j$:

$$\frac{1}{L_b}\|\boldsymbol{b}_j-C_{f^{(n+1)}(\boldsymbol{b}_j)}^{(n)}(\boldsymbol{b}_j)\|_2^2 \le \frac{1}{L_b}\|\boldsymbol{b}_j-C_{f^{(n)}(\boldsymbol{b}_j)}^{(n)}(\boldsymbol{b}_j)\|_2^2$$

That is, quantizing $\boldsymbol{b}_j$ at the end of the block clustering step of iteration $n+1$ results in lower quantization MSE compared to quantizing at the end of iteration $n$. Since this is true for all $\boldsymbol{b}\in\boldsymbol{X}$, we assert the following:

$$\tilde{J}^{(n+1)} = \frac{1}{N_c}\sum_{i=1}^{N_c}\frac{1}{|\mathcal{B}_i^{(n+1)}|}\sum_{\boldsymbol{b}\in\mathcal{B}_i^{(n+1)}}\frac{1}{L_b}\|\boldsymbol{b}-C_i^{(n)}(b)\|_2^2 \le J^{(n)} \tag{10}$$

where $\tilde{J}^{(n+1)}$ is the the quantization MSE after the vector clustering step at iteration $n+1$.

Next, during the codebook update step (6) at iteration $n+1$, the per-cluster codebooks $\mathcal{C}^{(n)}$ are updated to $\mathcal{C}^{(n+1)}$ by invoking the Lloyd-Max algorithm (Lloyd, 1982). We know that for any given value distribution, the Lloyd-Max algorithm minimizes the quantization MSE. Therefore, for a given vector cluster $\mathcal{B}_i$ we obtain the following result:

$$\frac{1}{|\mathcal{B}_i^{(n+1)}|}\sum_{\boldsymbol{b}\in\mathcal{B}_i^{(n+1)}}\frac{1}{L_b}\|\boldsymbol{b}-C_i^{(n+1)}(\boldsymbol{b})\|_2^2 \le \frac{1}{|\mathcal{B}_i^{(n+1)}|}\sum_{\boldsymbol{b}\in\mathcal{B}_i^{(n+1)}}\frac{1}{L_b}\|\boldsymbol{b}-C_i^{(n)}(\boldsymbol{b})\|_2^2 \tag{11}$$

The above equation states that quantizing the given block cluster $\mathcal{B}_i$ after updating the associated codebook from $C_i^{(n)}$ to $C_i^{(n+1)}$ results in lower quantization MSE. Since this is true for all the block clusters, we derive the following result:

$$J^{(n+1)} = \frac{1}{N_c}\sum_{i=1}^{N_c}\frac{1}{|\mathcal{B}_i^{(n+1)}|}\sum_{\boldsymbol{b}\in\mathcal{B}_i^{(n+1)}}\frac{1}{L_b}\|\boldsymbol{b}-C_i^{(n+1)}(\boldsymbol{b})\|_2^2 \le \tilde{J}^{(n+1)} \tag{12}$$

Following (10) and (12), we find that the quantization MSE is non-increasing for each iteration, that is, $J^{(1)} \ge J^{(2)} \ge J^{(3)} \ge \ldots \ge J^{(M)}$ where $M$ is the maximum number of iterations. ∎

Figure 9 shows the empirical convergence of LO-BCQ across several block lengths and number of codebooks. Also, the MSE achieved by LO-BCQ is compared to baselines such as MXFP and VSQ. As shown, LO-BCQ converges to a lower MSE than the baselines. Further, we achieve better convergence for larger number of codebooks ($N_c$) and for a smaller block length ($L_b$), both of which increase the bitwidth of BCQ (see Eq 3).

## A.3  Additional Accuracy Results

## A.4  Number Formats and Quantization Method

### A.4.1  Integer Format

An $n$-bit signed integer (INT) is typically represented with a 2s-complement format (Yao et al., 2022; Xiao et al., 2023; Dai et al., 2021), where the most significant bit denotes the sign.

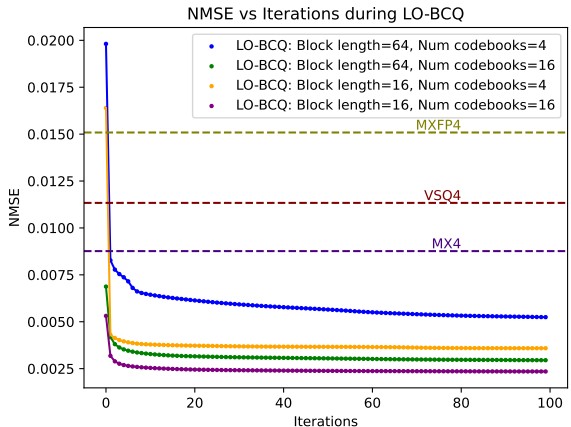

Figure 9: NMSE vs iterations during LO-BCQ compared to other block quantization proposals

| $L_b \rightarrow$ | 8 | | | | 4 | | 2 |
|---|---|---|---|---|---|---|---|
| $N_c$ / $L_A$ | 2 | 4 | 8 | 16 | 2 | 4 | 2 |
| GPT3-1.3B (FP32 PPL = 9.98) | | | | | | | |
| 64 | 10.40 | 10.23 | 10.17 | 10.15 | 10.28 | 10.18 | 10.19 |
| 32 | 10.25 | 10.20 | 10.15 | 10.12 | 10.23 | 10.17 | 10.17 |
| 16 | 10.22 | 10.16 | 10.10 | 10.09 | 10.21 | 10.14 | 10.16 |
| GPT3-8B (FP32 PPL = 7.38) | | | | | | | |
| 64 | 7.61 | 7.52 | 7.48 | 7.47 | 7.55 | 7.49 | 7.50 |
| 32 | 7.52 | 7.50 | 7.46 | 7.45 | 7.52 | 7.48 | 7.48 |
| 16 | 7.51 | 7.48 | 7.44 | 7.44 | 7.51 | 7.49 | 7.47 |

Table 12: Wikitext-103 perplexity across GPT3-1.3B and 8B models.

| $L_b \rightarrow$ | 8 | | | |
|---|---|---|---|---|
| $N_c$ / $L_A$ | 2 | 4 | 8 | 16 |
| Llama2-7B (FP32 PPL = 5.06) | | | | |
| 64 | 5.31 | 5.26 | 5.19 | 5.18 |
| 32 | 5.23 | 5.25 | 5.18 | 5.15 |
| 16 | 5.23 | 5.19 | 5.16 | 5.14 |
| Nemotron4-15B (FP32 PPL = 5.87) | | | | |
| 64 | 6.3 | 6.20 | 6.13 | 6.08 |
| 32 | 6.24 | 6.12 | 6.07 | 6.03 |
| 16 | 6.12 | 6.14 | 6.04 | 6.02 |
| Nemotron4-340B (FP32 PPL = 3.48) | | | | |
| 64 | 3.67 | 3.62 | 3.60 | 3.59 |
| 32 | 3.63 | 3.61 | 3.59 | 3.56 |
| 16 | 3.61 | 3.58 | 3.57 | 3.55 |

Table 13: Wikitext-103 perplexity compared to FP32 baseline in Llama2-7B and Nemotron4-15B, 340B models

| $L_b \rightarrow$ | 8 | | | | 8 | | | |
|---|---|---|---|---|---|---|---|---|
| $N_c$ / $L_A$ | 2 | 4 | 8 | 16 | 2 | 4 | 8 | 16 |
| Llama2-7B (FP32 Accuracy = 45.8%) | | | | | Llama2-70B (FP32 Accuracy = 69.12%) | | | |
| 64 | 43.9 | 43.4 | 43.9 | 44.9 | 68.07 | 68.27 | 68.17 | 68.75 |
| 32 | 44.5 | 43.8 | 44.9 | 44.5 | 68.37 | 68.51 | 68.35 | 68.27 |
| 16 | 43.9 | 42.7 | 44.9 | 45 | 68.12 | 68.77 | 68.31 | 68.59 |
| GPT3-22B (FP32 Accuracy = 38.75%) | | | | | Nemotron4-15B (FP32 Accuracy = 64.3%) | | | |
| 64 | 36.71 | 38.85 | 38.13 | 38.92 | 63.17 | 62.36 | 63.72 | 64.09 |
| 32 | 37.95 | 38.69 | 39.45 | 38.34 | 64.05 | 62.30 | 63.8 | 64.33 |
| 16 | 38.88 | 38.80 | 38.31 | 38.92 | 63.22 | 63.51 | 63.93 | 64.43 |

Table 14: Accuracy on MMLU dataset across GPT3-22B, Llama2-7B, 70B and Nemotron4-15B models.

| $L_b \rightarrow$ | 8 | | | | 8 | | | |
|---|---|---|---|---|---|---|---|---|
| $N_c$ / $L_A$ | 2 | 4 | 8 | 16 | 2 | 4 | 8 | 16 |
| Race (FP32 Accuracy = 37.51%) | | | | | Boolq (FP32 Accuracy = 64.62%) | | | |
| 64 | 36.94 | 37.13 | 36.27 | 37.13 | 63.73 | 62.26 | 63.49 | 63.36 |
| 32 | 37.03 | 36.36 | 36.08 | 37.03 | 62.54 | 63.51 | 63.49 | 63.55 |
| 16 | 37.03 | 37.03 | 36.46 | 37.03 | 61.1 | 63.79 | 63.58 | 63.33 |
| Winogrande (FP32 Accuracy = 58.01%) | | | | | Piqa (FP32 Accuracy = 74.21%) | | | |
| 64 | 58.17 | 57.22 | 57.85 | 58.33 | 73.01 | 73.07 | 73.07 | 72.80 |
| 32 | 59.12 | 58.09 | 57.85 | 58.41 | 73.01 | 73.94 | 72.74 | 73.18 |
| 16 | 57.93 | 58.88 | 57.93 | 58.56 | 73.94 | 72.80 | 73.01 | 73.94 |

Table 15: Accuracy on LM evaluation harness tasks on GPT3-1.3B model.

| $L_b \rightarrow$ | 8 | | | | 8 | | | |
|---|---|---|---|---|---|---|---|---|
| $N_c$ / $L_A$ | 2 | 4 | 8 | 16 | 2 | 4 | 8 | 16 |
| Race (FP32 Accuracy = 41.34%) | | | | | Boolq (FP32 Accuracy = 68.32%) | | | |
| 64 | 40.48 | 40.10 | 39.43 | 39.90 | 69.20 | 68.41 | 69.45 | 68.56 |
| 32 | 39.52 | 39.52 | 40.77 | 39.62 | 68.32 | 67.43 | 68.17 | 69.30 |
| 16 | 39.81 | 39.71 | 39.90 | 40.38 | 68.10 | 66.33 | 69.51 | 69.42 |
| Winogrande (FP32 Accuracy = 67.88%) | | | | | Piqa (FP32 Accuracy = 78.78%) | | | |
| 64 | 66.85 | 66.61 | 67.72 | 67.88 | 77.31 | 77.42 | 77.75 | 77.64 |
| 32 | 67.25 | 67.72 | 67.72 | 67.00 | 77.31 | 77.04 | 77.80 | 77.37 |
| 16 | 68.11 | 68.90 | 67.88 | 67.48 | 77.37 | 78.13 | 78.13 | 77.69 |

Table 16: Accuracy on LM evaluation harness tasks on GPT3-8B model.

| $L_b \rightarrow$ | 8 | | | | 8 | | | |
|---|---|---|---|---|---|---|---|---|
| $N_c$ $L_A$ | 2 | 4 | 8 | 16 | 2 | 4 | 8 | 16 |
| Race (FP32 Accuracy = 40.67%) | | | | | Boolq (FP32 Accuracy = 76.54%) | | | |
| 64 | 40.48 | 40.10 | 39.43 | 39.90 | 75.41 | 75.11 | 77.09 | 75.66 |
| 32 | 39.52 | 39.52 | 40.77 | 39.62 | 76.02 | 76.02 | 75.96 | 75.35 |
| 16 | 39.81 | 39.71 | 39.90 | 40.38 | 75.05 | 73.82 | 75.72 | 76.09 |
| Winogrande (FP32 Accuracy = 70.64%) | | | | | Piqa (FP32 Accuracy = 79.16%) | | | |
| 64 | 69.14 | 70.17 | 70.17 | 70.56 | 78.24 | 79.00 | 78.62 | 78.73 |
| 32 | 70.96 | 69.69 | 71.27 | 69.30 | 78.56 | 79.49 | 79.16 | 78.89 |
| 16 | 71.03 | 69.53 | 69.69 | 70.40 | 78.13 | 79.16 | 79.00 | 79.00 |

Table 17: Accuracy on LM evaluation harness tasks on GPT3-22B model.

| $L_b \rightarrow$ | 8 | | | | 8 | | | |
|---|---|---|---|---|---|---|---|---|
| $N_c$ $L_A$ | 2 | 4 | 8 | 16 | 2 | 4 | 8 | 16 |
| Race (FP32 Accuracy = 44.4%) | | | | | Boolq (FP32 Accuracy = 79.29%) | | | |
| 64 | 42.49 | 42.51 | 42.58 | 43.45 | 77.58 | 77.37 | 77.43 | 78.1 |
| 32 | 43.35 | 42.49 | 43.64 | 43.73 | 77.86 | 75.32 | 77.28 | 77.86 |
| 16 | 44.21 | 44.21 | 43.64 | 42.97 | 78.65 | 77 | 76.94 | 77.98 |
| Winogrande (FP32 Accuracy = 69.38%) | | | | | Piqa (FP32 Accuracy = 78.07%) | | | |
| 64 | 68.9 | 68.43 | 69.77 | 68.19 | 77.09 | 76.82 | 77.09 | 77.86 |
| 32 | 69.38 | 68.51 | 68.82 | 68.90 | 78.07 | 76.71 | 78.07 | 77.86 |
| 16 | 69.53 | 67.09 | 69.38 | 68.90 | 77.37 | 77.8 | 77.91 | 77.69 |

Table 18: Accuracy on LM evaluation harness tasks on Llama2-7B model.

| $L_b \rightarrow$ | 8 | | | | 8 | | | |
|---|---|---|---|---|---|---|---|---|
| $N_c$ $L_A$ | 2 | 4 | 8 | 16 | 2 | 4 | 8 | 16 |
| Race (FP32 Accuracy = 48.8%) | | | | | Boolq (FP32 Accuracy = 85.23%) | | | |
| 64 | 49.00 | 49.00 | 49.28 | 48.71 | 82.82 | 84.28 | 84.03 | 84.25 |
| 32 | 49.57 | 48.52 | 48.33 | 49.28 | 83.85 | 84.46 | 84.31 | 84.93 |
| 16 | 49.85 | 49.09 | 49.28 | 48.99 | 85.11 | 84.46 | 84.61 | 83.94 |
| Winogrande (FP32 Accuracy = 79.95%) | | | | | Piqa (FP32 Accuracy = 81.56%) | | | |
| 64 | 78.77 | 78.45 | 78.37 | 79.16 | 81.45 | 80.69 | 81.45 | 81.5 |
| 32 | 78.45 | 79.01 | 78.69 | 80.66 | 81.56 | 80.58 | 81.18 | 81.34 |
| 16 | 79.95 | 79.56 | 79.79 | 79.72 | 81.28 | 81.66 | 81.28 | 80.96 |

Table 19: Accuracy on LM evaluation harness tasks on Llama2-70B model.

### A.4.2 Floating Point Format

An $n$-bit signed floating point (FP) number $x$ comprises of a 1-bit sign ($x_{\text{sign}}$), $B_m$-bit mantissa ($x_{\text{mant}}$) and $B_e$-bit exponent ($x_{\text{exp}}$) such that $B_m + B_e = n - 1$. The associated constant exponent bias ($E_{\text{bias}}$) is computed as ($2^{B_e-1} - 1$). We denote this format as $E_{B_e} M_{B_m}$.

### A.4.3 Quantization Scheme

A quantization scheme dictates how a given unquantized tensor is converted to its quantized representation. We consider FP formats for the purpose of illustration. Given an unquantized tensor $\boldsymbol{X}$ and an FP format $E_{B_e} M_{B_m}$, we first, we compute the quantization scale factor $s_X$ that maps the maximum absolute value of $\boldsymbol{X}$ to the maximum quantization level of the $E_{B_e} M_{B_m}$ format as follows:

$$s_X = \frac{\max(|\boldsymbol{X}|)}{\max(E_{B_e} M_{B_m})} \tag{13}$$

In the above equation, $|\cdot|$ denotes the absolute value function.

Next, we scale $\boldsymbol{X}$ by $s_X$ and quantize it to $\hat{\boldsymbol{X}}$ by rounding it to the nearest quantization level of $E_{B_e} M_{B_m}$ as:

$$\hat{\boldsymbol{X}} = \text{round-to-nearest}\left(\frac{\boldsymbol{X}}{s_X}, E_{B_e} M_{B_m}\right) \tag{14}$$

We perform dynamic max-scaled quantization (Wu et al., 2020b), where the scale factor $s$ for activations is dynamically computed during runtime.

### A.5 Vector Scaled Quantization

During VSQ (Dai et al., 2021), the operand tensors are decomposed into 1D vectors in a hardware friendly manner as shown in Figure 10. Since the decomposed tensors are used as operands in matrix multiplications during inference, it is beneficial to perform this decomposition along the reduction dimension of the multiplication. The vectorwise quantization is performed similar to tensorwise quantization described in Equations 13 and 14, where a scale factor $s_v$ is required for each vector $\boldsymbol{v}$ that maps the maximum absolute value of that vector to the maximum quantization level. While smaller vector lengths can lead to larger accuracy gains, the associated memory and computational overheads due to the per-vector scale factors increases. To alleviate these overheads, VSQ (Dai et al., 2021) proposed a second level quantization of the per-vector scale factors to unsigned

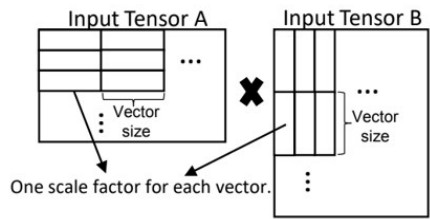

Figure 10: Vectorwise decomposition for per-vector scaled quantization (VSQ (Dai et al., 2021)).

integers, while MX (Rouhani et al., 2023b) quantizes them to integer powers of 2 (denoted as $2^{INT}$).

### A.5.1 MX Format

The MX format proposed in (Rouhani et al., 2023a) introduces the concept of sub-block shifting. For every two scalar elements of $b$-bits each, there is a shared exponent bit. The value of this exponent bit is determined through an empirical analysis that targets minimizing quantization MSE. We note that the FP format $E_1 M_b$ is strictly better than MX from an accuracy perspective since it allocates a dedicated exponent bit to each scalar as opposed to sharing it across two scalars. Therefore, we conservatively bound the accuracy of a $b + 2$-bit signed MX format with that of a $E_1 M_b$ format in our comparisons. For instance, we use E1M2 format as a proxy for MX4.

Figure 11 compares our 4-bit LO-BCQ block format to MX (Rouhani et al., 2023a). As shown, both LO-BCQ and MX decompose a given operand tensor into block arrays and each block array into blocks. Similar to MX,

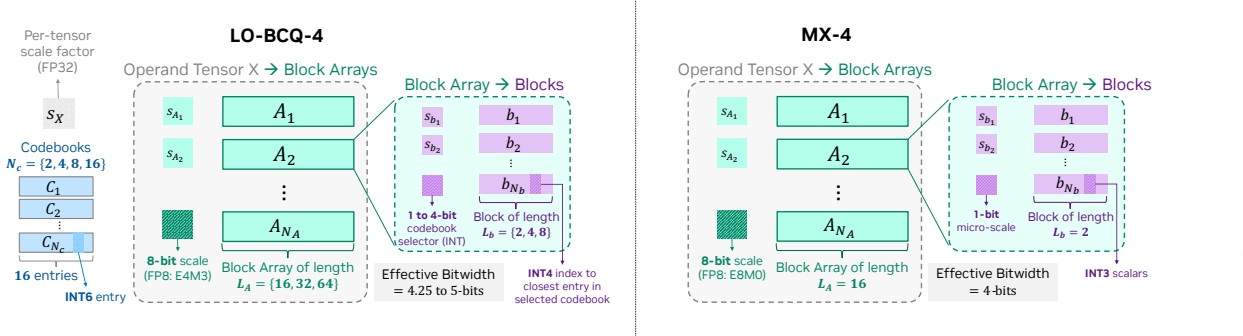

Figure 11: Comparing LO-BCQ to MX format.

we find that per-block quantization ($L_b < L_A$) leads to better accuracy due to increased flexibility. While MX achieves this through per-block 1-bit micro-scales, we associate a dedicated codebook to each block through a per-block codebook selector. Further, MX quantizes the per-block array scale-factor to E8M0 format without per-tensor scaling. In contrast during LO-BCQ, we find that per-tensor scaling combined with quantization of per-block array scale-factor to E4M3 format results in superior inference accuracy across models.

