# OpenReview forum: "LO-BCQ: Locally Optimal Block Clustered Quantization for 4-bit (W4A4) LLM Inference"
_TMLR — Accepted by TMLR_

### Review · Reviewer_bj5M · 2025-07-09

**Summary Of Contributions:**

The paper considers the problem of post-training quantization (PTQ) for large language models (LLMs) to 4-bits for both weights and activations (W4A4). The proposed method, Block Clustered Quantization (BCQ), first groups contiguous scalars in a tensor into blocks. It then clusters these blocks and assigns a dedicated codebook to each cluster for quantization. The paper introduces an iterative algorithm, Locally-Optimal BCQ (LO-BCQ) to jointly optimize the block-to-cluster assignments and the per-cluster codebooks by alternating between a block mapping step and a codebook generation step using the Lloyd-Max algorithm.

**Audience:**

Yes

**Claims And Evidence:**

Yes

**Requested Changes:**

N/A

**Strengths And Weaknesses:**

- This paper presents a well-executed and intuitive idea for weights and activation quantization that yields good empirical results.  The experimental evaluation is reasonable and largely supports the primary claims.

- The proposed idea builds on methods like MX4 and VSQ by allowing selection from multiple quantizers. This seems reasonable and logical step.

- Please consider discussing the choice of hyperparameters like codeword bitwidth, scale factor bitwidth and how they are selected.


Overall, I found the paper provide a simple and effective idea. Since my knowledge in the area is limited, I might not be able to fully appreciate the contextualization of this work in the literature.

---

> ### Author Response · Authors · 2025-07-21
>
> We sincerely thank the reviewer for their thoughtful and positive feedback on our work. We appreciate the acknowledgment of the clarity and effectiveness of the proposed idea and its empirical results.
>
> Regarding the hyperparameter choices, we selected the scale factor bitwidth as 8 bits in E4M3 format, aligning with the NVFP4 format used in recent GPUs [1]. For the codeword bitwidth, we conducted an ablation study on INT4, INT6, and INT8 codewords on Llama2-7B model and Wikitext-103 datatset. As shown in Table 1, LO-BCQ with INT6 achieves negligible perplexity degradation compared to INT8, while INT4 suffers significantly larger degradation. As a result, we quantize LO-BCQ codebooks to INT6. We will include these details and results in the final draft.
>
> **Table 1:** Quantizing LO-BCQ codebook entries to INT4 vs INT6 vs INT8 on Wikitext-103.
> Perplexity is measured on the Llama2-7B model (BF16 PPL = 5.06).
>
> | Codebooks         | INT4 | INT6 | INT8 |
> |-------------------|------|------|------|
> | g128, 2 codebooks | 6.24 | 5.38 | 5.35 |
> | g128, 4 codebooks | 6.21 | 5.27 | 5.25 |
> | g128, 8 codebooks | 6.20 | 5.21 | 5.21 |
> | g128, 16 codebooks| 6.18 | 5.21 | 5.19 |
>
> [1] https://developer.nvidia.com/blog/introducing-nvfp4-for-efficient-and-accurate-low-precision-inference

---

### Review · Reviewer_uJtg · 2025-07-13

**Summary Of Contributions:**

I found paper pretty challenging to read. Sections 2.1–2.2 do a good job of laying out the idea of block-clustered quantization, but they never say “codebook entries live in 6 bits” or “scale factors get 8 bits” until later.
As a result, readers naturally assume all quantized values use the same bit-width, Only in sec 3 (footnote 3) and sec 4.1 do they reveal. The bit width used in this paper all should be in section 2 clearly explained not in foot note.

Paper can be summarized by its core Algorithm (LO-BCQ)
1. Block to Codebook assignment
    * For each normalized block, compute average MSE versus each of the N codebooks (each with 16 FP32 centroids de-quantized from 6-bit storage).
    * Pick the codebook with the smallest FP32 MSE—store its selector index.
2. Codebook update
    * Gather all normalized scalars assigned to each cluster and run 1-D k-means to refine its 16 centroids.
    * Warm-start from previous centroids.


Novelty & Contributions
* Dynamic block clustering  lets each block pick the codebook that best fits its distribution.
* Alternating, locally-optimal assignment + centroid updates with k-means++.
* Two-level scale quantization so per-block dynamic ranges don’t blow up the bit-budget.
* Universal codebooks that generalize across layers and models, removing the need for per-tensor calibrations.
* End-to-end system demonstrating weight+activation quantization on LLMs with <1 % perplexity or accuracy loss.

**Audience:**

Yes

**Claims And Evidence:**

Yes

**Requested Changes:**

* Please clarify the bit width in the section 2, as the algorithm, and definition is explained in section 2.

**Strengths And Weaknesses:**

Strength :
* Strong Empricial results across different variants of LLM on different standard metrics.
* General two step approach to quantize any model.

Weakness :
* Paper is not clearly written, please refer to summary

I don't see much weakness as the authors use numbers to justify the method. Numbers look so good, I really wanted to try the quantize model to double check the evaluation score. If they could open source the code or provide the steps for evaluation that would help.

---

> ### Author Response · Authors · 2025-07-21
>
> We sincerely thank the reviewer for their valuable feedback and for highlighting the strength of our empirical results.
>
> We acknowledge the reviewer’s concern regarding the delayed introduction of bitwidth details for codebook entries and scale factors. We agree that presenting this information upfront would improve readability and avoid confusion. In the revised version, we will explicitly clarify in Section 2 that codebook entries are quantized to 6 bits and scale factors to 8 bits.
>
> As per the reviewer’s request, we detail our evaluation steps as follows. Please note that these steps are also discussed in Sections 3 and 4.
>
> -- We apply our LO-BCQ algorithm to operands before inference and pre-calibrate the optimal codebooks. We perform this calibration using one batch of activations from the GPT3-126M model on Wikitext-103 dataset. We freeze these optimal codebooks across operands and models.
>
> -- During inference, we quantize each operand block by mapping it to a codebook that best represents it, i.e, the one that minimizes quantization MSE.  Subsequently, we perform quantized GEMM operations using these mapped representations.

---

### Review · Reviewer_4WbB · 2025-08-11

**Summary Of Contributions:**

Recent work has sought to improve the performance of quantized neural network by quantizing operands by blocks, allowing for different quantization configurations for different parts of a model. The common approach is to optimize a set of code books such that the error of quantization for the blocks is minimal. Here, the authors propose improvements to the approach, where they also optimize the assignment of blocks to codebooks. By doing this, they develop a new clustering+quantization algorithm that performs block clustering with state-of-the-art performance on compressing large language models at 4-bit weights and activation (W4A4) precision.

**Audience:**

Yes

**Claims And Evidence:**

Yes

**Requested Changes:**

I believe the work could be strengthened by further discussing the hyperparameter tuning. In particular, further discussion on the effect of and motivations for block size selection would be interesting. For example, why not have a block for every vector, so that every vector can be assigned independently to a codebook?

I would also be curious to know more about the properties of codebooks and blocks learned by the authors' method. Ie. is it possible to analyze how they are quantizing operands differently from previous approaches?

Finally, an analysis of the BCQ approach on tasks outside of the domain of NLP would be valuable. However, I acknowledge that the scope of the paper is explicitly LLMs.

In general, I think the paper is well-written and self-contained with convincing results as is, so I do not think any of the above suggestions are necessary for my recommendation for acceptance.

**Strengths And Weaknesses:**

Strengths:
- The paper is well written, and provides good background information on block quantization methods needed to understand the authors' approach
- The approach is elegant and explained clearly: both the assignment of blocks to codebooks and codebook optimization happens in an alternating scheme, inspired by existing approaches to minimize MSE and using the LloydMax algorithm
- The approach is benchmarked with large Transformer models, showcasing its robustness, applicability, and benefits at scale

Weaknesses:
- Most of the results are related to language modelling. Large-scale models are used for a variety of tasks outside of language, so while I would still expect it to work well, ultimately the authors' approach has not been tested as extensively on different areas such as vision tasks, where people are also interested in quantization. It could be that BCQ in general is a strong quantization approach
- The work does not significantly explore and discuss the choice of hyperparameters (number of clusters/codebooks) beyond the evaluation of a few configurations
- There is a lack of discussion about how the resultant quantization produces similar or different results from other approaches. If there are interesting features learned by the approach, it would be interesting to see an analysis of them.

---

> ### Author Response · Authors · 2025-08-22
>
> We sincerely thank the reviewer for the insightful questions and suggestions.
>
> We fully agree with the reviewer that a more detailed discussion of parameter choices—particularly block size and the number of codebooks—would add clarity to our method. In our study, we systematically explored the design space under a bitwidth constraint between 4 and 5 bits. As described in Equation (9) of the paper, the effective bitwidth is determined by a combination of factors, including block size, block array size, number of codebooks, and the bitwidth of per-block array scale factors and codebook entries. Importantly, if we were to adopt a design where each block had its own independent codebook, the bitwidth overhead required to store the additional codebooks would be significant. This overhead would quickly dominate the bitwidth budget, making such a design impractical under the constraints we target. For completeness, the Table 9 in the Appendix of the paper provides a comprehensive sweep of possible configurations and their corresponding bitwidths. In the final version of the paper, we will include a more explicit discussion on LO-BCQ parameter selection, as suggested.
>
> We are very grateful for the reviewer’s suggestion to further analyze the nature of the learned codebooks. Conventional quantization methods typically use fixed number formats, such as integers or floating-point representations, where the spacing between quantization levels follows a predetermined and rigid pattern. In contrast, our BCQ approach leverages codebooks that are free from such constraints. This is further detailed in Figure 6 in the paper where the codebooks identified by LO-BCQ are compared against number formats such as INT4(E3M0) and FP4(E2M1 and E1M2). As shown, codebooks identified by LO-BCQ captures the inherent asymmetric and non-uniform value distributions in LLMs to minimize the mean-squared error during quantization.
>
> We greatly appreciate the reviewer’s suggestion to extend the analysis of BCQ approach beyond NLP. While the scope of our experimental evaluations in this paper was LLMs, we would like to emphasize that our algorithmic approach that minimizes quantization MSE is applicable to any given value distribution. Indeed, in Appendix Section A.2, we provide a theoretical justification of this generality. We will explore the extension of our approach to non-NLP domains in future work.

---

### Decision · Action_Editor_KFvt · 2025-09-26

**Recommendation:** Accept as is

**Audience:**

Yes

**Audience Explanation:**

The proposed Block Clustered Quantization is an interesting and novel PTQ method.

**Claims And Evidence:**

Yes

**Claims Explanation:**

The work has provided strong empirical results and has received recognition from reviewers.